# Revisiting Retinal Degeneration Hallmarks: Insights from Molecular Markers and Therapy Perspectives

**DOI:** 10.3390/ijms241713079

**Published:** 2023-08-23

**Authors:** João Gabriel Santos Rosa, Geonildo Rodrigo Disner, Felipe Justiniano Pinto, Carla Lima, Monica Lopes-Ferreira

**Affiliations:** Immunoregulation Unit, Laboratory of Applied Toxinology (CeTICs/FAPESP), Butantan Institute, São Paulo 05503900, Brazil; joao.rosa.esib@esib.butantan.gov.br (J.G.S.R.); geonildo.disner.esib@esib.butantan.gov.br (G.R.D.); felipe.pinto.esib@esib.butantan.gov.br (F.J.P.); carla.lima@butantan.gov.br (C.L.)

**Keywords:** eye health, visual impairment, age-related macular degeneration, glaucoma, retinitis pigmentosa, diabetic retinopathy, therapeutic strategies

## Abstract

Visual impairment and blindness are a growing public health problem as they reduce the life quality of millions of people. The management and treatment of these diseases represent scientific and therapeutic challenges because different cellular and molecular actors involved in the pathophysiology are still being identified. Visual system components, particularly retinal cells, are extremely sensitive to genetic or metabolic alterations, and immune responses activated by local insults contribute to biological events, culminating in vision loss and irreversible blindness. Several ocular diseases are linked to retinal cell loss, and some of them, such as retinitis pigmentosa, age-related macular degeneration, glaucoma, and diabetic retinopathy, are characterized by pathophysiological hallmarks that represent possibilities to study and develop novel treatments for retinal cell degeneration. Here, we present a compilation of revisited information on retinal degeneration, including pathophysiological and molecular features and biochemical hallmarks, and possible research directions for novel treatments to assist as a guide for innovative research. The knowledge expansion upon the mechanistic bases of the pathobiology of eye diseases, including information on complex interactions of genetic predisposition, chronic inflammation, and environmental and aging-related factors, will prompt the identification of new therapeutic strategies.

## 1. Introduction

The eyes are paramount highly specialized sensory organs that are capable of receiving visual images, which account for around 75% of the information captured around us. The eyes are one of the most complex structures in the human body and one of the most prone to downturns [1]. Eye evolution is driven by the accumulation of gradually more demanding behaviors that continuously increase performance requirements of photoreceptor organs, notably, through the rise of a light-dependent chemical reaction coupled to a signaling system [2]. The phototransduction system and vision neurophysiology enable complex behaviors in response to information, marking one of the most distinguishable features of metazoans. Environmental factors and diseases can affect both optical and neurophysiological components of the vision system, reflecting and predicting important pathologies.

Visual impairment and blindness are a growing public health problem as they reduce the quality of life of millions of people worldwide. First, they represent scientific and therapeutic challenges because different cellular and molecular actors involved in the pathophysiology are still being identified. Furthermore, they represent a public health challenge as they require all countries to prioritize a global eye health policy [3].

Eyes consist of three distinguishable layers, which include the optically clear aqueous humor, lens, and vitreous body [4]. The outermost layer consists of the cornea and sclera; the intermediary layer contains the main blood supply, consisting of the choroid, ciliary body, the iris; while the innermost layer is the retina, which receives most of the nourishment from vessels within the choroid and converts light to chemical energy. The neurons of the neural retina translate visual information into nerve impulses, which travel through the optic nerve to the brain [4]. Overall, eyes are generously vascularized, ensuring a stable temperature and preventing changes in retinal blood flow over a wide range of perfusion pressures [5].

Thus, the whole retinal ecosystem formed by rod and cone photoreceptors, supportive cells, and the retinal pigment epithelium (RPE) is extremely sensitive to genetic or metabolic alterations, and immune cells activated by insults contribute and lead to vision impairments and, ultimately, vision loss or irreversible blindness [6].

Several ocular diseases are linked to retinal cell loss, and diseases such as retinitis pigmentosa (RP), age-related macular degeneration (AMD), and diabetic retinopathy (DR) are characterized by photoreceptor degeneration; however, optic neuropathies, such as glaucoma, also lead to vision loss owing to retinal ganglion cell (RGC) loss [7].

Rod and cone photoreceptors are highly specialized glutamate-releasing neurons that have a high metabolic status because they detect and transmit visual information through an intense cellular process known as phototransduction. In this process, photons reach photosensitive opsin proteins, culminating with photoreceptor hyperpolarization and a decrease in glutamate, which triggers signaling between retinal cells [8].

The conversion of visual information from the retina to the brain is the responsibility of RGCs, a type of neuron located near the inner surface (the ganglion cell layer) [9].

RGCs may present diversity in terms of types of association, size, and reaction to visual stimuli. They receive visual information from photoreceptors via two types of intermediary neurons: bipolar cells, a type of glutamatergic neuron, and amacrine cells, a type of nondopaminergic neuron. Bipolar cells help transmit light signals from photoreceptors to ganglion cells, while amacrine cells are necessary for the development of functional units on the inner surface of the retina. Thus, inner retinal cells—amacrine, bipolar, and horizontal cells (inhibitory interneurons that locally modulate the photoreceptor synaptic output)—can process, modulate, and integrate information from photoreceptors to RGCs. From RGCs, ganglion cell axons emerge and form the optic nerve, which transmits information to optic centers in the brain [10].

The RPE is a photoreceptor-adjacent cell layer that acts as a physiological mediator between the posterior segment of the eye and the choroid, which also forms the blood–retinal barrier (BRB). RPE performs metabolic control of photoreceptors and preserves retina homeostasis, assuring normal visual function [11].

Here, we present a compilation of revisited information on retinal degeneration, including pathophysiological and molecular features and biochemical hallmarks, and possible directions for novel treatments to assist as a guide for innovative research. Understanding these retinal degeneration processes and their different hallmarks will facilitate novel prospects and the design of new therapeutic strategies. Toward this objective, we searched the PubMed database for information on diseases related to damage to posterior structures of the eye, such as age-related macular degeneration (AMD), retinal ganglion cell degeneration (Glaucoma), retinitis pigmentosa (RP), and diabetic retinopathy (DR)—Figure 1—and selected publications from the past five years, including reputable information in older publications.

## 2. Selected Retinal Degeneration Diseases

### 2.1. Age-Related Macular Degeneration (AMD)

AMD is characterized by the degeneration of photoreceptors and is a multifactorial disease that involves genetic, aging-related, and environmental factors. This visual disturbance accounts for approximately 9% of blindness and is the most common cause of blindness in elderly people.

According to Wong et al. (2014), by 2040, 288 million people will have AMD [12]. Modern lifestyle and habits, such as smoking and poor diets, are risk factors for developing AMD. As an example of genetic factors, mutations in genes *cfh* and *arms2*–*htra1* [13] confer the highest risk of AMD. Retinal disease that present chronic inflammation and inadequate retinal extra-cellular matrix (ECM) maintenance, with increased lipid and lipoprotein deposition, are often associated with oxidative stress [14].

Moreover, drusen (yellow–white structures containing cholesterol, proteins, and extracellular debris) and subretinal drusenoid deposits are common risk factors for AMD development. Klein et al. (2007) found a positive relation between age and druse size, indicating that elderly patients with late-stage AMD present larger drusen throughout disease development [15]. Thus, druse size is clinically used to classify the intensity of AMD, wherein small and medium drusen indicate early-stage AMD, while the largest drusen suggest late-stage AMD [16,17]. Another hallmark of advanced AMD is geographic atrophy (GA)—a severe alteration in tissue morphology caused by photoreceptor, RPE, and choriocapillaris atrophy—which results in progressive vision loss, especially in the central visual field [18]. Accordingly, large drusen associated with pigment changes are robust indicators of advanced AMD, GA, and/or neovascularization, a condition also called *dry AMD* [16]. 

AMD pathophysiology can present as macular neovascularization (MNV), which is considered as a hallmark of neovascular AMD [19]. The aggravation of neovascular AMD is characterized by the aggressive development of choroidal neovascularization, rupture of nonfunctional vessels, and extravasation of fluid into intra- and subretinal compartments (*wet AMD*).

Defective treatment of *dry AMD* and exudative *wet AMD* conditions results in fibrosis and severe vision loss [14]. Both variants equally lead to RPE destruction and consequent loss of photoreceptors and visual capacity failure, mainly of central vision (reviewed in [20]). Several pharmacological approaches are used to treat *wet AMD* once it is a chronic and multifactorial disease. The classic pharmacological treatment is the use of repeated intraocular doses of the monoclonal antibody against the vascular endothelial growth factor (anti-VEGF) over a prolonged period, often for life. Endophthalmitis (inflammation within the eye), retinal detachment, cataracts, and increased intraocular pressure are known to occur in the context of intravitreal therapy.

As reviewed recently, the treatment for concomitant systemic diseases with drugs, such as immunosuppressants, cholesterol-reducing agents, nonsteroidal anti-inflammatory drugs, dopamine precursors, and hypoglycemic agents, might impact the disease course and offer variable degrees of protection and/or regression. Some examples include anti-VEGF agents that have longer durations of action, ankyrin repeat protein (DARPin)-based therapy that binds all VEGF isoforms, bispecific anti-VEGF/angiopoietin (Ang)-2 therapies, anti-PDGF and anti-integrin therapies, Rho-kinase inhibitors, the port delivery system, steroids, gene therapies for the retina and uveitis and glaucoma, RhO-kinase (ROCK) inhibitors, implants and plugs, selective laser trabeculoplasty, and minimally invasive glaucoma surgeries [21,22].

Oxidative stress is the disbalance between the production of reactive oxygen species (ROS) from cellular processes and the antioxidant defense system [23]; and highly metabolically active cells, such as photoreceptors, are constantly exposed to ROS, which makes them susceptible to damage by oxidative stress [24]. Among cellular functions of the RPE are light absorption, transport of nutrients, and phagocytosis of shed photoreceptor outer segments. During periods of high metabolic activity, photoreceptor phagocytosis increases, and the ROS concentration elevates, followed by RPE degeneration. Thus, this dysfunction in the macular region leads to photoreceptor degeneration, which contributes to AMD progression [25]. 

Bruch’s membrane (BM) rests between the RPE and choriocapillaris and is under the constant effect of matrix metalloproteinases (MMPs) and tissue inhibitors of metalloproteinases (TIMPs) to maintain ECM homeostasis. Changes in BM and the disruption of the ECM seem to be involved in AMD progression. Age-related changes in BM include accumulation of cellular debris, lipid deposition, and alteration in the ECM [26]. Alterations in the RPE and BM seem to be initial signals of AMD once structural changes occur owing to lipoprotein accumulation [27]. Rhodopsin metabolization produces all-trans retinol, and its reaction with lipids and proteins generates a pigment called lipofuscin, for which accumulation leads to RPE cell death and atrophy in addition to activation of retinoic acid receptor genes and VEGF expression, which increase the risk of choroidal neovascularization.

#### Therapeutic Approaches

A number of studies have revisited altered parameters of metabolites in AMD patients and described altered lipid, amino acid, and glucose metabolisms, which indicate that the metabolomic profile aids the discovery of different pathophysiological features and novel treatments (reviewed in [28]). The cellular response to aggression engages in AMD progression. However, the imbalance between MMPs and TIMPs provokes alteration in elastin and collagen compositions in BM, affecting the extracellular matrix structure and function [27]. The VEGF pathway plays a significant role in AMD pathogenesis, and extreme situations, such as photoreceptor hypoxia or oxidative stress, can trigger VEGF activity.

The goal of AMD therapeutics is to improve vision prognosis through retinal cell recovery, and the injection of pharmacological substances represents a robust strategy for AMD management. Anti-VEGF medicines have been used in a consolidative manner to achieve success, and the first drug approved in the United States was pegaptanib, which selectively binds and neutralizes VEGF 165 [29]. 

Bevacizumab is a humanized monoclonal antibody that is approved for cancer treatment; however, it is suitable for AMD treatment as an off-label drug. Intravitreal injection is the usual administration form, and bevacizumab acts by binding and neutralizing VEGF-A [30]. In addition, ranibizumab is a monoclonal antibody fragment, which is used to neutralize all VEGF-A isoforms [31]. 

Moreover, faricimab is a bispecific antibody that is also used for AMD treatment. This antibody is an efficient VEGF-A and Angiopoietin-3 neutralizer and is successfully used to treat AMD over longer intervals, which improves patients’ quality of life [32]. The rationale in the use of anti-VEGF drugs is to prolong visual ability and contain lesions, and newer drugs, such ranibizumab, have been developed to have a more durable effect through surgical implantation of drug delivery systems [30]. In addition to humanized antibodies, different molecules can be used as therapeutic agents, such as abicipar, a VEGF-A binding protein [33]. Complement inhibitors also have important therapeutic approaches. Recently, C3 and C5 inhibitors pegcetacoplan and avacincaptad pegol, respectively, were reported to be effective in controlling AMD [34]. 

### 2.2. Retinal Ganglion Cell Degeneration (Glaucoma)

Glaucoma is a heterogeneous group of eye disorders that cause the loss of RGCs, and the most prevalent form is primary open-angle glaucoma (POAG) [35,36]. Glaucoma is the second leading cause of blindness worldwide after cataracts and mainly worsens when not diagnosed and treated early [9]. Since 2020, the number of people with glaucoma has exceeded 80 million globally; and, as more effective diagnostic tools become available, the number of cases is projected to increase [37], potentially reaching 111 million by 2040.

The primary cause of glaucoma is not known, but this condition is usually engendered by fluid building up in the front part of the eye, which increases the pressure. Moreover, a plethora of risk factors has been identified, including age, genetic predisposition, multiple genes, individual risk factors, and environmental elements, which are likely to contribute to the disease onset [9]. A previous report (reviewed in [36]) indicated that glaucomatous optic neuropathy (GON), also known as the pathohistological feature of glaucoma in the optic nerve, either originates from compromised mechanical conditions at the lamina cribrosa or is associated with pathological vascular involvement [38]. Such impairment initially occurs in the laminar area and is linked with several factors, such as the disruption of the neurotrophic factor, glial activation, release of tumor necrosis factor (TNF), oxidative stress, dysregulation of the immune system, and mitochondrial dysfunction [39,40,41,42].

#### 2.2.1. Glaucoma Hallmarks and Genetic Basis

The understanding of the background covering the molecular basis of glaucoma has been of great interest in science for an extended period. For decades, researchers have turned toward genetics to better understand the cause of glaucoma. Owing to improvements in and increasing accessibility to genomic technology, it has been possible to cover an extensive genetic basis of individuals who are or are not affected to determine which specific gene loads and mutations play a role in the disease.

The most recognized marker indicative of symptoms leading to RGC degeneration is (1) elevated intraocular pressure (IOP), also referred to as ocular hypertension [43]. It occurs when the fluid pressure inside the eye is too elevated. However, not all people with ocular hypertension develop glaucoma and vice versa [44]. In fact, the IOP is currently the only modifiable disease feature because neuroprotective therapies are unavailable. Nevertheless, these treatments are not restorative; they just seek to slow the disease progression. Sadly, more than half of glaucoma diagnoses take place when irreversible optic nerve damage has already occurred [45].

Available treatments, including prostaglandin analogs, carbonic-anhydrase inhibitors, β-adrenergic antagonists, α2-adrenergic agonists, and Rho-kinase inhibitors, are usually effective at lowering the IOP and controlling the disease progression. However, many patients do not reach a satisfactory IOP, and at least one-third make use of combinatory therapy with multiple IOP-lowering drugs that have complementary mechanisms; supporting treatments may even include laser treatment and surgery to help the fluid drain [43,46,47]. Recent findings indicate that even low intracranial pressures can also be a risk factor for the development of normal-tension glaucoma. Thus, the higher the translaminar pressure difference (TPD) is, the more significant the visual-field damage will be [48].

Other essential factors in glaucoma are (2) optic nerve damage and (3) visual field loss [44,49]. The optic nerve conducts visual information from the eye to the brain, and its deterioration (by poor blood flow or genetic abnormalities, for example) can cause vision loss. On the other hand, visual field loss is the result of glaucoma progression that can start as the loss of peripheral vision that, over time, can become more severe and lead to blindness.

Glaucoma can (4) also cause structural changes in the eye, noticeably, shape and size, or even the appearance of the optic nerve head, which can be easily diagnosed through eye exams and imaging tests or visual-field testing. Moreover, other complex multifactorial risk factors are considered important hallmarks of glaucoma. Those include (5) age, family history, and ethnicity as well as certain medical conditions, such as diabetes, hypertension, and nearsightedness [49]. As the cells constituting the eye become more prone to harm with time, glaucoma is more common in older adults and people with a family history of the disease and could be worsened by other comorbidities. However, in reference to ancestry, there is still some conflicting research ongoing. Documented research has found ethnicity as a risk factor for early and advanced loss archetypes, with people of African descent being at higher risk for developing glaucoma [50].

The reasons for the increased susceptibility in African or Latino descendants are not fully understood, but several factors may play a role. Among those elements are genetic factors (i.e., genes may be involved in regulating the pressure inside the eye or the function of the optic nerve); structural differences in the eye (e.g., a thinner cornea or larger optic nerve head, which can affect the accuracy of intraocular pressure measurements and increase the risk for developing glaucoma); and environmental factors (such as diet, exposure to toxins, and socioeconomic factors).

However, the first analysis of glaucoma in multiple ancestries from the largest genome-wide association study of glaucoma (GWAS) to date revealed that the majority of known risk loci for POAG have been identified in European, Asian, and African ancestries [45], contradicting observational studies that indicate ethnicity-related prevalence. The same study used a dataset of more than 34,000 adults with glaucoma to describe 127 genes associated with the condition by identifying 44 new gene loci and confirming 83 previously reported loci linked to glaucoma. The integration of multiple lines of genetic evidence supports the functional relevance of the described glaucoma risk loci and highlights potential contributions of several genes to pathogenesis, including *svep1*, *rere*, *vcam1*, *znf638*, *clic5*, *slc2a12*, *yap1*, *mxra5*, and *smad6*.

Although age is a risk factor that is well described for the increase in vision loss due to glaucoma, recent research has demonstrated a new genetic mutation behind childhood glaucoma that may be a root cause of a severe condition affecting children’s vision by the age of three [35]. Through advanced genome-sequencing technology, a mutation in the thrombospondin-1 (*thbs1*) gene was found in three ethnically and geographically diverse families with childhood glaucoma histories. Additionally, the findings were confirmed in a mouse model presenting the genetic mutation. The authors identified the heterozygous gene *thbs1*, a missense allele that alters p.Arg1034, a highly evolutionarily conserved amino acid, which affects congenital glaucoma, especially among children.

Several other genes have been identified in association with an increased risk of glaucoma, including the myocilin gene (*myoc*), a commonly mutated gene associated with the most common form of glaucoma (POAG) (reviewed in [51,52,53]). In addition, between 10 and 30% of individuals with juvenile open-angle glaucoma have mutations in the gene that encodes myocilin [9]. Its relevance is that mutations in this gene can interfere with the intraocular pressure, damage the optic nerve, and alter the aqueous humor dynamics [54].

The optineurin gene (*optn*) is another gene that is associated with POAG, second to *myoc*. Mutations and haplotype variants have been found in some people with early-onset POAG and may be regarded as potential contributing factors to primary glaucoma [53,55]. Additionally, He et al. (2019) found an association between the *optn* T34T variant and normal-tension glaucoma (NTG), indicating that this gene might be implicated in the disease through a mechanism not related to ocular pressure increase [56].

Mutations in the WD repeat domain 36 gene (*wdr36*) may also affect the function of proteins involved in regulating the IOP and cause severe retinal damage mainly by impairing RGC axon growth [57,58]. Curiously, when Chi et al. (2010) investigated a mutant *wdr36* gene expressed in all mice tissues, just the defects in the retina could be portrayed [58]. Parallelly, in previous studies using zebrafish (*Danio rerio*) to determine the function of *wdr36* (the homolog of human *wdr36*), Skarie and Link (2008) have shown developmental defects after the loss of the *wdr36* function, including a small head and eyes with lens opacity and thickening of the lens epithelium but relatively mild defects in the retina (even at six months post fertilization) [59].

Further genes that have been linked to glaucoma include the cytochrome P450, family 1, subfamily B, polypeptide 1 gene (*cyp1b1*), which is notably associated with congenital glaucoma—a rare form of glaucoma that manifests at birth or within the first few years of life [9,60,61,62], and the cyclin-dependent kinase inhibitor 2B antisense RNA 1 gene (*cdkn2b-as1*) [63,64] among others.

Zhao et al. (2022) reported that the ubiquitous protein sigma 1 receptor, which is well-known to protect cells from stress, appears to have a key function for ensuring the survival of RGCs in vitro. In experiments wherein RGCs and astrocytes were cultured together in a dish, both cell types survived, unless the astrocytes were missing their sigma 1 receptor [65]. The study also provided some of the first evidence that drugs that activate sigma 1 receptors, such as the pain reliever pentazocine, may one day help mitigate damage from glaucoma once it reduces the generation of potentially destructive ROS and protects astrocytes from death. Likewise, sigma 1 receptor activation increases the activity of the synapses on the optic nerve head, including an increase in STAT3, which plays an essential role in many cellular functions and is known to regulate the reactivity of astrocytes.

Similarly, Zhu et al. (2013) had already suggested the role of hypoxia-inducible factor-1α (*hif-1α*) in the preconditioning-induced protection of RGC [66]. The group demonstrated in a mouse model that endogenous mechanisms can be activated by a repetitive hypoxic preconditioning (RHP) stimulus to provide consistent RGC protection. In mutated mice lacking *hif-1α* in RGCs, the results corroborate that the transcription factor exerted a protective function from glaucomatous injury.

Interestingly, new research [67] reveals the role of apolipoprotein E4 (*apoe4*), a genetic variant associated with Alzheimer’s disease, in protecting against glaucoma. The study found that in two mouse glaucoma models, microglia transitioned to a neurodegenerative phenotype characterized by the upregulation of *apoe* and *lgals3* (Galectin-3). Mice with targeted deletion of *apoe* in microglia or carrying the human *apoe4* allele were protected from RGC loss, despite elevated IOPs. These results demonstrate that the impaired activation of *apoe4* microglia is protective against glaucoma and that APOE-Galectin-3 signaling can be targeted to treat this disease.

Another relevant aspect of glaucoma is the sequence of biochemical events triggered by the alteration in the expression of different elements connected to the regulation of cellular oxidative stress and homeostasis. Observations indicate that before degeneration, hypoxic stress could be the initial stress. A notorious example is thioredoxins, small redox proteins that function as antioxidants by facilitating the reduction of other proteins. Munemasa et al. (2009) observed decreased thioredoxin 1 (*trx1*) and thioredoxin 2 (*trx2*) levels in the glaucomatous retina, and overexpression of these proteins supports RGC survival [68]. Even IOP elevation induces oxidative stress in RGCs through decreased activity of several enzymes comprising the antioxidant defense system, including superoxide dismutase, glutathione peroxidase, and catalase, which have been implicated in RGC body death [69].

The retinal glia-mediated inflammatory response plays a critical role in RGC death in glaucoma. TNF-α and interleukin (IL)-1β cytokines, produced by activated glial cells, may promote gliosis and the inflammatory response of activated Müller cells, thus aggravating RGC injury in glaucoma [39]. In the glaucomatous retina, activated glial cells contribute to cellular death by releasing inflammatory signals. Furthermore, nitric oxide (NO) synthase 2 (*nos2*), expressed in the presence of cytokines, when in high concentrations, can be neurotoxic. Moreover, endoplasmic reticulum (ER) stress can also induce RGC degeneration, accompanied by increased ER stress-related proteins, such as Bip, PERK, and CHOP [70]. The amplified expression of ER stress-related proteins is detrimental to the retina, and ER stress plays an important role in retinal cellular apoptosis [71].

#### 2.2.2. Therapeutic Approaches in Glaucoma Research

Glaucoma is commonly known as the “silent thief of sight”, as it remains asymptomatic until later stages, and, consequently, its diagnosis is delayed [72]. There are treatments to delay vision loss but no cure, making it a leading cause of irreversible blindness worldwide. Accordingly, further knowledge of the disease pathophysiology is urgent to help in the diagnosis and development of new and effective treatment strategies because currently applied medical therapies are limited and may cause adverse side effects.

Despite the clinical heterogeneity of glaucoma, IOP remains the only treatable factor. Topical glaucoma medications decrease the IOP by reducing aqueous humor production or improving outflow. There is accumulating evidence that NO plays a major role in IOP control through direct action on the trabecular meshwork and, hence, lowers IOP [73]. An increasing number of NO donors have been developed for glaucoma and ocular hypertension treatment. Merged therapies can induce synergistic effects on IOP decrease, such as NO-donating β-blockers, NO-donating prostaglandins, NO-donating carbonic anhydrase inhibitors, and the dual NO donor delivery system [74].

Moreover, a great deal of drugs targeting glaucoma risk genes may be potential therapeutic candidates. The number of identified molecular risk factors can lead to the discovery of new biological pathways and, consequently, putative targets. Therefore, gene therapy for retinal ganglion cell neuroprotection in glaucoma has been considered as an alternative method of treatment for over a decade [9]. Notably, with the advent of viral and nonviral agents suitable for in vivo gene delivery, gene therapy has gained considerable ground [75].

Among available viral vector systems, the adeno-associated virus (AAV) vector has emerged as a preferable tool for targeting RGCs. The ability of AAV vectors to transduce distinct retinal cell types depends on the virus serotype, the route of vector administration, and the age of the host [76,77]. DNA- and RNA-based technologies are also of great benefit for modifying gene expression. DNA plasmids or oligonucleotides are easy to work with and can be readily injected into eyes, but they are not easily taken up by cells, which may result in just slight protection of axotomized RGCs owing to limited transfection efficiency [77]. Small interference RNA (siRNA) has been successfully delivered to RGCs via injection into the superior colliculus; however, the highly invasive nature of this approach limits its clinical application [78].

Using gene-editing systems, scientists have developed new models of glaucoma, which resemble primary congenital glaucoma, in mice. By injecting a new, long-lasting, and nontoxic protein treatment (Hepta-ANGPT1) into mice, they were able to replace the function of genes that, when mutated, cause glaucoma. This same treatment, when injected into the eyes of healthy adult mice, reduced pressure in the eyes, which supports this treatment as a possible new class of therapy for the most common cause of glaucoma in adults [47].

In addition, for putative molecules that might contribute to neuroprotection to prevent vision loss, a study found that activating the calcium/calmodulin-dependent protein kinase II (CaMKII) pathway aids to protect RGCs from a variety of injuries in multiple mouse glaucoma models [79]. However, depending on the conditions, the CaMKII activity inhibition is either protective or detrimental to RGCs. Using an antibody marker of CaMKII activity, the authors identified that this signaling pathway was compromised whenever RGCs were exposed to toxins or traumatic injury to the optic nerve, suggesting a correlation between CaMKII activity and retinal ganglion cell survival. By looking for intervention strategies, the researchers found that activating the CaMKII pathway via genetic modification proved to be protective to cells.

Providing gene treatment to mice just prior to the toxic insult and just after the optic-nerve crush increased the CaMKII activity and robustly protected RGCs. Among gene-therapy-treated mice, 77% of RGCs survived one year after the toxic insult compared to 8% in the control. Six months following the optic-nerve crush, 77% of RGCs survived, while only 7% survived in controls. Correspondingly, boosting the CaMKII activity was also effective in glaucoma models according to elevated eye pressures or genetic deficiencies [79].

Furthermore, it was demonstrated that Copaxone 1, a compound used in the treatment of multiple sclerosis, inhibits RGC loss after the optic-nerve crush [80], indicating that the modulation of the autoimmune response is a relevant research direction toward the development of novel strategies for neuroprotection. Neuroprotective therapies would be a leap forward, meeting the needs of patients who lack treatment options. Furthermore, axonal protection has been indicated as a therapeutic strategy in the prevention of preperimetric glaucoma [36].

Finally, there is a medical arsenal of other agents that are routinely used in the treatment of some secondary glaucomas, such as corticosteroids, anticholinergics, and anti-VEGFs, as adjunctive therapies in the management of neovascular glaucoma [81].

### 2.3. Retinitis Pigmentosa (RP)

Degenerative retinal diseases, such as RP, are within the scope of inherited retinal dystrophies (IRDs), and several genes are shared between RP and other IRDs; however, mutations in around 80 genes are believed to be involved exclusively in RP, comprising more than 3000 mutations. These genetic alterations can be transmitted in an autosomal-recessive, autosomal-dominant, or X-linked manner, generating diverse phenotypes (reviewed in [82]). In this manner, X-linked RP provokes severe clinical signs, while autosomal dominant RP presents mild visual impairment. However, despite most patients being declared legally blind owing to vision loss, some maintain a degree of visual ability because macular function is preserved (reviewed in [83]).

Regardless of the heterogeneous phenotypes of RP, rod degeneration in the early stages of RP leads to peripheral vision loss, with preserved visual acuity, once the macular function is preserved at this stage of the disease; yet, this so-called “tunnel vision” declines, as the disease progresses, to cone loss. Photoreceptor degeneration is a hallmark of RP, especially rod photoreceptor death before cone photoreceptor degeneration in the most advanced stages of the disease.

Clinically, RP is a progressive disease, gradually impairing night vision and visual acuity [6]. Photoreceptor functions can be assessed using electroretinograms, which show abnormal results and, along with clinical manifestations and complementary exams, such as optical coherence tomography—assessment of the macular morphology in a noninvasive way—are robust callsigns of RP. Morphologically, the outer nuclear layer of the retina, formed by photoreceptor nuclei, is affected, presenting a diminished aspect owing to cellular degeneration. The support cells (amacrine, horizontal, and bipolar) remain preserved until the late stages of the disease [6].

According to its pathology, RP covers several syndromic and nonsyndromic disorders. About 20 to 30% of RP patients present the syndromic form, with clinical signs associated with extra-ocular abnormalities. In these syndromes, in addition to vision loss, hearing loss and vestibular dysfunction occur in Usher syndrome, and polydactyly, genital abnormality, cognitive impairment, and classic RP symptoms are common features of Bardet–Biedl syndrome, which is an autosomal-recessive hereditary disease caused by *bbs1-bbs21* gene mutations [84]. Interestingly, in two specific types of syndromic RP, it is possible to preserve vision with clinical treatment. Supplementation of vitamin A and vitamin E impairs the progression of retinal degeneration in Bassen–Kornzweig syndrome, whereas the dietary restriction of phytanic acid exerts the same effect on Refsum disease (reviewed in [83]).

Nonsyndromic RP is caused by a biochemical dysfunction that affects photoreceptor homeostasis, without the involvement of other organs, that could derive from light damage, apoptosis, ciliary transport dysfunction, and endoplasmic reticulum stress [85]. This form of RP has a worldwide prevalence of one in 5000 [86] and is considered as the most common IRD.

#### 2.3.1. Genetic Background

Although RP is a genetically heterogeneous disease, most cases of RP are monogenic, and mutations in the rhodopsin gene (*rho*), Usher syndrome 2A gene (*ush2a*), and retinitis pigmentosa GTPase regulator gene (*rpgr*) engage in about 30% of all cases of retinitis pigmentosa. Mutated genes encode nonfunctional proteins that interfere with biochemical pathways through the rod phototransduction cascade, producing classic clinical signs of RP (reviewed in [6]).

The first genetic cause of RP was described in 1990 by Dryja et al., who identified a nucleotide substitution at position 23 of the rod opsin protein [87]. The *rho* gene encodes the protein rhodopsin, a visual pigment necessary for normal vision, especially under low-light conditions, and essential for rod function and survival, as has been demonstrated in rhodopsin (Rho) knock-out (KO) mice that developed intense photoreceptor degeneration [88].

This photopigment is composed of a protein—opsin—and contains a nonprotein domain called 11-cis-retinal that absorbs light and turns into all-trans-retinal to initiate the phototransduction cascade. *Rho* mutations lead to misfolding opsin and are a quite common cause of autosomal-dominant RP. Recently, Parain et al. (2022) produced in both *Xenopus laevis* and *Xenopus tropicalis*, a mutation of the *rho* gene, successfully generating amphibian RP models to study the involvement of the Muller glia cell response in RP pathogenesis [89]. Since the Dryja study, several *rho* mutations have been described, and more than 150 mutations have been associated with the RP phenotype. Cellular pathways that are essential to rod maintenance are targets of *rho* mutations, which lead to pathological events, such as endocytosis dysfunction, structural instability of the OS, and disturbed protein traffic and folding. These mutations are dispersed across rhodopsin and may occur in the N-terminus region or transmembrane helices (reviewed in [88]).

Eventually, gene mutations activate the macromolecular aggregation or misfolding of proteins, a leading cause of cellular disfunction. Protein aggregation or misfolding leads to endoplasmic reticulum stress and triggers the unfolded protein response (UPR) that can cause cellular death through apoptotic pathways, such as caspase activation. This pathological mechanism has been described in RP, where misfolded proteorhodopsins, resulting from *rho* mutation, provoked intense protein aggregation, culminating in UPR activation and consequent photoreceptor degeneration [90].

Thus, the cellular protein homeostasis in the photoreceptor is altered, resulting in cellular death mechanism activation. In this process, the most common causes are (1) apoptosis, due to both caspase pathway activity and caspase-independent apoptosis, leading to both oxidative stress augmentation and accumulation of oxidative DNA damage. However, in P23H and S334ter mutants, nonapoptotic markers of cellular death were identified, which suggested increased oxidative DNA damage (reviewed in [88]).

Different cellular pathological processes participate in RP photoreceptor degeneration, such as (2) ER disfunction and consequent oxidative damage, as demonstrated in rhodopsin mutants T17M and E349X, where increased proinflammatory markers were associated with the ER stress response. Photoreceptor-programmed necrosis (3) could be mediated by receptor-interacting protein kinases (RIP), and inflammasome activation implicates cellular death in a P23H-1 rat model (reviewed in [88]).

FAM161A is a ciliary protein that is expressed in photoreceptor inner segments and plexiform layers and on retinal ganglion cells. FAM161A is a part of a cytoskeleton preservation system in microtubule-organizing centers and is encoded by the *fam161a* gene (reviewed in [91]). These proteins are essential for vision physiology and Beryozkin et al. (2021), using a *fam161a* KO (*fam161a*^tm1b/tm1b^) mouse, successfully modeled retinal degeneration, emphasizing the phenotypic resemblance between *fam161a* KO and human RP [92].

Photoreceptors have a very active metabolic environment, in which an intense traffic of signaling molecules and cellular processes are transported between inner and outer segments through the connective cilium, and several proteins regulate this process, including RPGR. This protein is encoded by the *rpgr* gene [93]. Mutations in *rpgr* compromise the transporter function of the cilium, ultimately leading to cellular death. Thus, genetic ablation of an *rpgr* gene constitutes a robust model of the human RP phenotype [94,95].

Usher syndrome (USH) is a disorder characterized by the loss of hearing and sight and represents about 50% of all hereditary deafness–blindness cases, with a prevalence from 4.4 to 16.6 per 100,000 people worldwide. Clinical manifestations comprise moderate to severe hearing loss from birth combined with vision impairment and gradual deterioration of rod photoreceptors—the RP hallmarks [96,97]. USH has complex characteristics because it can be subdivided into three types according to the genetic basis and corresponding clinical aspects. USH type 1 (USH1) causes congenital deafness, vestibular defects, and early-onset RP, and the *ush1c* gene is involved in USH1. Schäfer et al. (2023) demonstrated that one protein encoded by *ush1c*, the scaffold protein harmonin, interacts with coactivator β-catenin and suppresses the cWnt pathway, an important cell–cell communication pathway [98].

Mutation of the *ush2a* gene is the cause of Usher syndrome type 2, the most common form of USH. The *ush2a* gene encodes two isoforms of usherins, essential proteins for cochlear hair cell development and the maintenance of photoreceptors [99]. Moreover, mutations in *ush2a* can also produce autosomal-recessive RP without hearing involvement [97].

The *rpe65* gene encodes all-trans retinyl ester isomerase, an enzyme that is critical to the visual cycle, and biallelic mutations of this gene cause a severe and sight-threatening autosomal-recessive genetic disorder that leads to a severe form of rod–cone mediated IRD [100]. Individuals with *rpe65* biallelic mutations may receive one of a variety of clinical diagnoses, and the disease course may include early- or late-onset nystagmus along with night blindness and vision loss.

#### 2.3.2. Therapeutic Approach

Because the altered traffic or folding of rhodopsin represents a significant part of RP pathogenesis, chaperone molecules are considered as chemical substances with pharmacological activity and may be used as therapeutic agents [101]. Athanasiou et al. (2018) reviewed the use of substrate-specific chemical chaperones that improve rhodopsin stabilization and prevent protein aggregation [88].

The most recent RP therapies involve the use of genetic tools to inhibit the causal agent and supply exogenous protein [102]. The early retinal degeneration of RP has been treated since 2018 with gene therapy, Luxturna^®^, authorized by the American and European drug agencies (FDA—Food and Drug Administration and EMA—European Medicines Agency, respectively). Patients with Leber’s congenital amaurosis [103] and retinitis pigmentosa associated with biallelic bimutations of the *rpe65* gene can now receive this gene replacement in addition to other classes of drugs capable of maintaining a more stable retinal metabolic environment, such as neurotrophic factors, anti-apoptotic agents, and antioxidants [90,104].

### 2.4. Diabetic Retinopathy (DR)

The abnormal proliferation or function of blood vessels in the retina, which allows the extravasation of fluids and lipids into the central retina or macula, characterizes DR. Multifactorial microvascular complications accompanied by neurodegeneration and diabetic macular edema (DME) trigger a traction force on the surface of the retina, which leads to retinal detachment and loss of vision. DR is observed in patients with types I and II diabetes mellitus [105,106,107,108].

Diabetes mellitus occurs when there is a defective secretion of insulin by the pancreas. Although type I is characterized by the immune-mediated destruction of B cells, type II is marked by a resistance to or deficiency in insulin signaling. The 10th edition of the International Diabetes Federation’s, Diabetes Atlas (2021) points out that diabetes is the fastest-growing disease of the 21st century, with an incredible 783 million cases of diabetes worldwide predicted for 2045 [109].

DR affects about 93 million people of all ages worldwide and is the main cause of blindness or vision loss in working-age adults. It is further estimated that one in three diabetics will develop DR and that one in ten people with DR will have vision-threatening progression of the disease [110,111,112]. All insulin-dependent diabetic individuals will develop some type of complication resulting from the disease, whether macrovascular or microvascular, as in the case of DR, within 15 years after the initial diagnosis [113].

The main risk factors associated with diabetic retinopathy are the maintenance of the diabetic condition for a prolonged period without glycemic control, hypertension, dyslipidemia, obesity, ethnic origin, pregnancy, puberty, cataract surgery, and smoking. For example, a 1% reduction in the concentration of glycated hemoglobin (HbA1c) reduces the risk for developing retinopathies by 40%, and a 10 mmHg decrease in blood pressure reduces the risk of vision loss by 50% [114,115].

For a long time, DR was considered as just a retinal microvascular alteration, including capillary degeneration, pericyte loss, and vascular leakage, but studies carried out between 2010 and 2011 began to point out new bases for the development of the pathology, which came to be considered as a neurovascular disease [116]. Injuries to the retinas of diabetics were first observed in 1856, and were graded on a severity scale more than a century later, allowing the treatment of patients with DR to be more effective and the disease to be studied more clearly [117].

The classification of diabetic retinopathy can be divided into nonproliferative (NPDR), mild and moderate cases in the initial phase of the disease, and proliferative, severe cases characterized by increased vascular permeability, capillary occlusion, microaneurysms, hemorrhages, and hard exudates. It is noteworthy that asymptomatic patients do not present any symptoms and that in proliferative (PDR), which is a severe stage of DR, there is neovascularization, hypoxia, and vitreous hemorrhaging, and there may also be retinal detachment [118,119]. The disruption of the blood–retinal barrier is the most common cause of vision loss in DR because of swelling and thickening of the macula from the accumulation of fluid in the retina. Diabetic macular edema (DME) may occur in patients with or without NPDR or PDR and impacts visual perception [118,120]. The neovascularization observed in DR accompanied by fibrosis results in retinal detachment, which resolves only with surgery [121].

The development mechanisms of diabetic retinopathy are similar to those of chronic inflammatory diseases, such as increased vascular permeability, infiltration of proinflammatory cells, edema formation, tissue damage, neovascularization, and the production of cytokines and chemokines. Furthermore, a study carried out by Lutty (2013) confirmed the participation of inflammatory molecules in changes in the structure and function of retinal tissue in retinopathy [122].

#### 2.4.1. Molecular and Genetic Factors of Diabetic Retinopathy

Over many years, researchers have postulated several pathways or mechanisms that seem to be involved in the development and progression of severe cases of DR. With the increase in research carried out with DNA and RNA that has grown over the last twenty years, studies have begun to highlight the main genes and molecules involved in DR development.

First, among the main established mechanisms are the increase in the flow of the polyol pathway, increase in ROS, reduction in NO, and increase in the action of the enzyme aldose reductase. As a consequence, there is an increase in oxidative stress and de novo synthesis of diacylglycerol (DAG), which activates the abnormal signaling cascade of protein kinase C (PKC), important for regulating vascular permeability, angiogenesis, and inflammation [123]. Furthermore, there is an increase in the formation of advanced nonenzymatic glycosylation products (AGEs), proteins, or glycated lipids owing to the presence of large amounts of oxidized sugars, such as *N*^ε^-(carboxymethyl)lysine (CML) [124], increased stress, increased flow in the hexosamine pathway, and peripheral neurodegeneration.

Changes in neuronal cells, such as astrocytes, have been demonstrated in diabetic mice, and have progressed to internal hypoxia in the retina and Müller glial dysfunction, the latter being due to the accumulation of AGEs and advanced lipid-oxidation end products (ALEs) [125,126]. Furthermore, increases in insulin-like growth factor 1 and hypoxia-inducible factor 1 alpha in serum and vitreae are also observed. Upregulation causes hypoxia and local and systemic inflammation with sustained high levels of retinal inflammatory molecules, including VEGF, monocyte chemoattractant protein-1 (MCP-1), inducible nitric oxide synthase (iNOS), cyclooxygenase-2 (COX-2), IL-1β, and NF-κB [108].

In summary, hyperglycemia induced by diabetes mellitus triggers a series of biochemical reactions involving various pathways, including the activation of the polyol pathway, generation of advanced glycation end products, activation of protein kinase C, initiation of the hexosamine pathway, and regulation of polymerase. This imbalance results in dysfunctions within mitochondria, an upsurge in reactive oxygen species, the release of proinflammatory mediators, and hypoxia, ultimately leading to neurovascular damage and an increase in VEGF expression. Furthermore, emerging evidence suggests the direct involvement of the renin–angiotensin–aldosterone system by the observed dysfunction of neurovascular tissue [127].

In addition, activated Müller cells increase the production of VEGF and FCFb. These angiogenic mediators, in turn, promote the formation of fragile retinal blood vessels (i.e., retinal neovascularization), heralding the development of proliferative DR (PDR) [108].

Emerging evidence suggests that cellular senescence (cessation of cellular division) in the retina may contribute to the development of DR [128,129]. DR can continue to progress, even with strict glycemic control. Among the reasons for this progression is an important role of mitochondrial DNA (mtDNA), induced by oxidative stress, that has been pointed out in previous studies [130,131,132]. The cGAS/STING signaling pathway regulates both cellular senescence and inflammation and has been reported by BBB as a link between these processes during the pathogenesis of DR [133].

The risk of severe DR is about three-fold higher in siblings of affected individuals than in the general population [134], which is supported by findings from identical-twin studies [135]. Many genes have been identified that directly or indirectly act upon DR, such as angiotensin I converting enzyme (*ace*) [136]; angiotensin II receptor type 1 (*agtr1*) [137]; aldose reductase (*alr2*) [138]; endothelial nitric oxide synthase (*enos*) [139]; glucose transporter 1 (*glut1*) [140,141]; vascular endothelial growth factor (*vegf*) [142]; receptor for advanced glycation end products (*rage*) [143]; interleukin 1 beta (IL-1β); alpha-2 macroglobulin; complement components C1, C3, C2, and C1 inhibitor; angiotensinogen (*agt*) [144]; paraoxonase 1 (*pon1*) [145]; transforming growth factor beta (*tgf-β*) [146]; angiotensin converter and plasminogen activator inhibitor 1 (*pai*) [136]; and methylenetetrahydrofolate reductase (*mthfr*) [147] genes. All these studies were carried out using animal models, such as mice, and samples obtained from different populations worldwide. Some polymorphisms in regulatory regions were also identified and may contribute to the susceptibility and progression of diabetic DR [107].

A GWAS developed by Graham et al. (2018) identified *nrxn3* loci that have a probable association with PDR [120] and confirmed *pcks2* and *malrd1* loci in Caucasians, which had already been described by Grassi et al. (2011) [148]. In addition, the authors pointed out that some RNAs have also been related to pathology, as is the case of *linc00343* and *loc285626*.

In Mexican Americans, *camk4* and *fmn1* were identified as genes involved with DR [149]. In the Japanese population, a study by Awata et al. (2014) identified that the lincRNA locus RP1-90L14.1, which is close to the *cep162* gene, is associated with DR [150]. In Chinese, another study points to *tbc1d4-commd6-uchl3* and *lrp2-bbs5* and *arl4c-sh3bp4* loci [151]. Furthermore, susceptibility to DR was demonstrated for *plxdc2* and *arhgap22* genes in the Taiwanese population [152].

Moreover, microRNAs have already been implicated in glucose homeostasis processes and angiogenesis, such as miR-17-5p, miR-18a, miR-20a, miR-21, miR-31, and miR-155; the modulation of the inflammatory response, such as miR-146, miR-155, miR-132, and miR-21; and responses of diabetic retinopathy and diabetes in general [116].

#### 2.4.2. Therapeutic Approach

Recent findings in molecular and genetic factors involved in diabetic retinopathy are important for developing new biomarkers and more effective treatments for the disease. Currently, in addition to glycemic control, which has already proven effective for reducing the risk of developing DR in the long term for a portion of diabetic patients [153,154], other therapies are recommended for DR, such as the use of intravitreal anti-VEGF drugs (ranibizumab, bevacizumab, and aflibercept), which have the ability to inhibit angiogenesis in the retina by binding to VEGF-A and, thus, preventing the activation of the VEGFR2 receptor and hampering the formation of exudative blood vessels and the impairment of the retinal structure, as observed in diabetic retinopathy [121,155].

Laser photocoagulation treatment, applied directly to areas of retinal thickening, is also used to reduce the risk of vision loss. A clinical trial conducted in patients with DME showed a 17% decrease in this risk. However, it is essential to consider the potential risks associated with this technique, including burns, scar expansion, momentary increases in DME, and neovascularization [156].

Furthermore, intravitreal corticosteroids, namely, dexamethasone and fluocinolone acetonide, are authorized by the Food and Drug Administration (FDA) for therapy in patients with DME. Corticosteroids act on the inflammatory process by modifying the expression of proteins and shaping the immune response, leading to a decrease in the expression of VEGF. However, similar to anti-VEGF treatments, the intravitreal route poses the main challenge for this therapy, as it may potentially trigger glaucoma [157].

Several clinical trials are underway with other drugs for RD therapy, including squalamine (an angiogenic inhibitor), AKB-9778 (vascular permeability decreasing), nesvacumab (vascular permeability decreasing), and a bispecific antibody, RO6867461 (angiogenic and vascular permeability) [118]. Furthermore, a recent study pointed out that exosomes can be used for the treatment of diabetic retinopathy, such as miR-141-3p and miR-203a-3p, which can inhibit retinal neovascularization [158].

## 3. Conclusions

Eye health is essential to achieve overall health and wellbeing, social inclusion, and quality of life. Ocular diseases related to damage to the posterior structures of the eyes, such as age-related macular degeneration, retinal ganglion cell degeneration, retinitis pigmentosa, and diabetic retinopathy, are among the most frequent causes of visual impairment (596 million) and blindness (almost 45 million) in the adult population worldwide. Projections estimate an increase in global prevalence in the coming years owing to the higher incidence of metabolic diseases and increased life expectancy.

Minimally invasive surgeries, chemical molecules directed to protein targets or receptors, enzyme inhibitors, signaling-pathway inhibitors, and DNA- and RNA-based technologies constitute the foundation for the medical management of ocular diseases (Figure 2). However, expansion in the knowledge of mechanistic bases of the pathobiology of eye diseases, including information on the complex interactions of genetic predisposition, chronic inflammation, and environmental and aging-related factors, will allow the identification of new therapeutic strategies, applied locally or systemically, capable of curbing inflammation and senescence by suppressing inflammatory mediators and cells, inducing gene replacement, promoting neuroprotection, and maintaining the stability of the retinal metabolic and structural environments.

Over the past decade, significant advancements have been made in the treatment of retinal degeneration, bringing hope and improved outcomes for patients affected by these devastating medical conditions. A noteworthy development in retinal degeneration treatment is the advent of gene therapies. By understanding specific genetic mutations associated with inherited retinal diseases, researchers have successfully developed gene replacement therapies to restore or augment the expression of functional proteins in affected retinal cells. In the absence of effective or accessible alternatives to manage retinopathies, the facilitation of access to diagnostic testing is extremely valuable for identifying groups of people or individuals who are the most vulnerable. Additionally, early diagnosis is beneficial because routine eye exams are a crucial effort to safeguard vision. Early detection and treatment are essential as they might avert vision loss and blindness. Advancements in retinal imaging technologies have played a pivotal role for enhancing the early diagnosis and monitoring of retinal degeneration. They have enabled the visualization of retinal layers and cellular structures in unprecedented detail, aiding the identification of disease biomarkers and providing valuable insights into disease progression. Furthermore, imaging-guided treatment strategies have improved patient outcomes by allowing the precise delivery of therapeutic agents.

In conclusion, this comprehensive review sheds light on the key hallmarks of retinal degeneration and elucidates the intricate pathogenesis and implicated molecular biomarkers. Therapeutic perspectives underscore potential avenues for combating this debilitating medical condition by exploring strategies from gene and molecular therapies and monoclonal antibody interventions to the investigation of novel pharmacological targets. Therefore, there is a promising outlook for halting, or even reversing, retinal degeneration. However, it is crucial to acknowledge the complexity of this disease and the need for continued research collaboration and clinical trials to translate these promising perspectives into effective treatments for patients. By understanding the underlying mechanisms, identifying robust molecular biomarkers, and advancing innovative therapies, we may be one step closer to alleviating the burden of retinal degeneration and improving the quality of life for those affected.

## Figures and Tables

**Figure 1 ijms-24-13079-f001:**
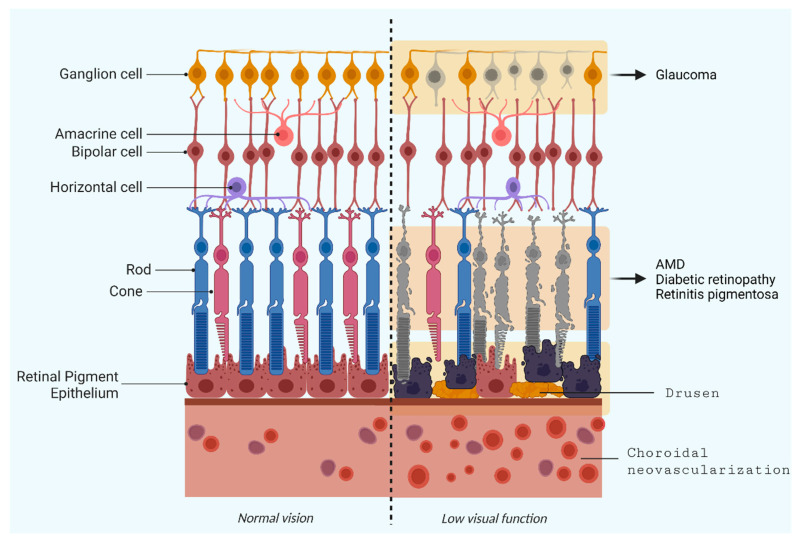
Schematic overview of retinal organization showing its main constituents and highlighting the cellular composition in normal vision (**left**) and under affected phenotypes (**right**), specifically, retinal ganglion cell degeneration (Glaucoma), age-related macular degeneration (AMD), diabetic retinopathy, and retinitis pigmentosa (RP).

**Figure 2 ijms-24-13079-f002:**
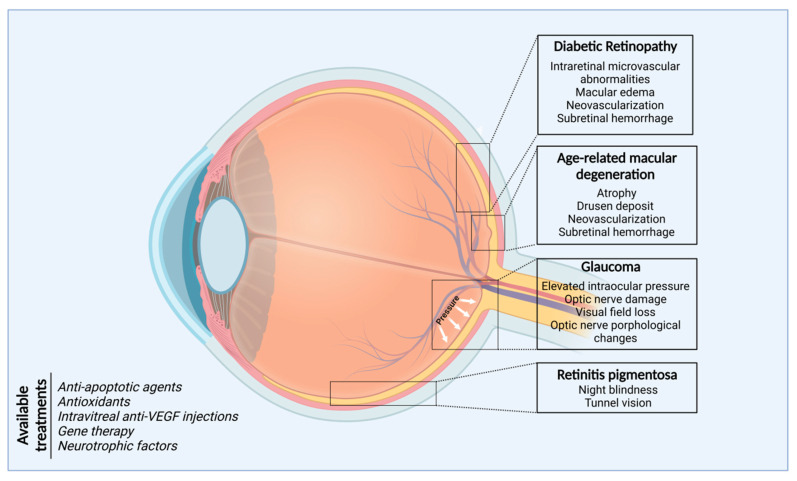
Hallmarks of the selected retinopathies and the most common treatments for the management of ocular diseases.

## Data Availability

Not applicable.

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
