# Peer review of "Revisiting Retinal Degeneration Hallmarks: Insights from Molecular Markers and Therapy Perspectives"

_ijms, 2023, doi:10.3390/ijms241713079_

Round 1

Reviewer 1 Report

While the review paper addresses an interesting issue, it is not accepted in its current form. There are several issues that need to be addressed. I encourage the authors to revise their manuscript and resubmit it to the journal again. 

Quality of the English should be improved; there were several language mistakes detected. 

Author Response

Reviewer #1

The paper is not well written and structured in my opinion. Even though the title of the paper is good but the contents or information provided throughout the paper is not enough to support the title. The paper has many shortcomings in regards to information, lack of many references, language mistakes, too many segmentations or paragraphs under each of the topic starting from introduction. Given these shortcomings the manuscript requires major revisions. I encourage the authors to revise their manuscript and to resubmit it to the journal. In its current form, this paper cannot be considered for publication.

Comments to the authors:

In the introduction enough information have not been provided. It needs major improvement. There are several language mistakes, therefore, I recommend a professional round of language editing before the paper is published.

Answer: the authors appreciate your willingness to review the paper and your valuable recommendations.

Line 34-39: No references provided throughout.

A: The proper reference was added to this text excerpt.

“References lacking at many places which is the major issues detected throughout the paper”

A: References were inserted in the text.

The figures provided has no information/legends. Even denoting Fig 1 to the first figure is lacking.

A: The figures were corrected.

Starting with age-related macular degeneration, no abbreviations used. It would have much better to provide some figures related to it. Same goes for other pathologies.

A: Abbreviations in the text were standardized and a list of abbreviations was inserted.

For AMD, there were no separate paragraphs or topic related to causes as well therapies but for glaucoma and RP separate paragraphs and topic or subheading were provided for it causes and therapies. This has to be consistent for all the diseases mentioned in the paper.

A: The text was reorganized according to reviewer’s suggestions. The sections were segmented to include a therapeutic approach for each disease.

Line 422: 5 e 6??

A: The proper reference was added to this text excerpt.

Conclusions is very short.

A: The conclusion session was better elaborated.

The figures provided has no information/legends. Even denoting Fig 2 to the second figure is lacking.

A: The figures were corrected.

Reviewer 2 Report

The authors reviewed a series of studies related to ophthalmic retinal diseases, such as retinitis pigmentosa, age-related macular degeneration, glaucoma, and diabetic retinopathy, and presented an introduction to these latest treatment modalities. This information will be useful to researchers who are to start their research in these areas and will guide them in the direction to proceed. However, I have identified some points that may be misinterpreted by the reader and suggest some corrections.

1) In this process, photons reach photosensitive opsin proteins, culminating with PR membrane potential alteration and release of neurotransmitters, (page3, line50)

> Actually, photons induce hyperpolarization of the membrane and “stop” the release of glutamate from PR. Please rephrase the sentence.

2) The number of RGCs varies between individuals and may present diversity in terms of types of association, size and reaction to visual stimuli. (page3, line55)

> Are there any references that have shown the RGC variation between individuals?

3) and amacrine cell, a dopaminergic neuron. (page3, line 58)

> Amacrine cells include non-dopaminergic cells, such as GABAergic, glycinergic and cholinergic cells.

4) In Figure 1, “Retinitis pigmentosa” should be added to the ONL row in the right column because the photoreceptor degeneration could be induced by gene mutation itself.

5) Parain et al. (2022) produced in both, Xenopus laevis and Xenopus tropicalis, a mutation on rho gene, successfully generating RP models to study the involvement of Muller glia cells response on RP pathogenesis [75]. (page13, line445)

> Although it is well known that Müller cells are involved in retinal regeneration in amphibians and fishes, their involvement in regeneration in mammals is still under debate.

Author Response

Reviewer #2

The authors reviewed a series of studies related to ophthalmic retinal diseases, such as retinitis pigmentosa, age-related macular degeneration, glaucoma, and diabetic retinopathy, and presented an introduction to these latest treatment modalities. This information will be useful to researchers who are to start their research in these areas and will guide them in the direction to proceed. However, I have identified some points that may be misinterpreted by the reader and suggest some corrections.

Answer: the authors appreciate your willingness to review the paper and your valuable recommendations.

 1) In this process, photons reach photosensitive opsin proteins, culminating with PR membrane potential alteration and release of neurotransmitters, (page3, line50)

> Actually, photons induce hyperpolarization of the membrane and “stop” the release of glutamate from PR. Please rephrase the sentence.

A: The sentence was corrected according to reviewer indication.

2) The number of RGCs varies between individuals and may present diversity in terms of types of association, size and reaction to visual stimuli. (page3, line55)

> Are there any references that have shown the RGC variation between individuals?

A: The sentence was corrected according to reviewer indication.

3) and amacrine cell, a dopaminergic neuron. (page3, line 58)

> Amacrine cells include non-dopaminergic cells, such as GABAergic, glycinergic and cholinergic cells.

A: The sentence was corrected according to reviewer indication.

4) In Figure 1, “Retinitis pigmentosa” should be added to the ONL row in the right column because the photoreceptor degeneration could be induced by gene mutation itself.

A: The figure was corrected.

5) Parain et al. (2022) produced in both, Xenopus laevis and Xenopus tropicalis, a mutation on rho gene, successfully generating RP models to study the involvement of Muller glia cells response on RP pathogenesis [75]. (page13, line445)

 > Although it is well known that Müller cells are involved in retinal regeneration in amphibians and fishes, their involvement in regeneration in mammals is still under debate.

A: The word “amphibian” was inserted in the sentence, evidencing the application of the RP model.

Reviewer 3 Report

The review article by Joao Gabriel Santos Rosa et al compiles the recent developments in retinal degeneration studies with emphasis on age related macular degeneration, glaucoma, retinitis pigmentosa and diabetic retinopathy. This would be a contribution to understand the pathophysiology and recent developments in the field of degenerative retinal disorder research. The authors did a detailed literature survey and pointed out the important developments in the manuscript. Here are some observations to improve the article before acceptance.

1.     Retinitis pigmentosa is not detailed in the article. Over 150 different mutations of  Rhodopsin have been identifies for RP and it would be good if the authors mention some of those mutants here and about the misfolding/aggregation mechanisms.

Author Response

Reviewer #3

The review article by Joao Gabriel Santos Rosa et al compiles the recent developments in retinal degeneration studies with emphasis on age related macular degeneration, glaucoma, retinitis pigmentosa and diabetic retinopathy. This would be a contribution to understand the pathophysiology and recent developments in the field of degenerative retinal disorder research. The authors did a detailed literature survey and pointed out the important developments in the manuscript. Here are some observations to improve the article before acceptance.

Answer: the authors appreciate your willingness to review the paper and your valuable recommendations.

  1. Retinitis pigmentosa is not detailed in the article. Over 150 different mutations of  Rhodopsin have been identifies for RP and it would be good if the authors mention some of those mutants here and about the misfolding/aggregation mechanisms.

A: The follow text was inserted:

“ Since Dryja study, several rho mutations have been described, and more than 150 mutations are associated with RP phenotype. Cellular pathways that are essential to rods maintenance are target of rho mutations, which mutations lead to pathological events such as endocytosis dysfunction, structural instability of the OS, and disturbed protein traffic and folding. These mutations are dispersed across rhodopsin and may occur in N-terminus region, or in transmembrane helices (reviewed in [82]).

Eventually, gene mutations activate macromolecular aggregation or misfolding of proteins, a leading cause of cellular disfunction. The protein aggregation or misfolding leads to endoplasmic reticulum stress and trigger the unfolded protein response (UPR) that can cause cell death through apoptotic pathways, such as caspase activation. This pathological mechanism has been described in RP, where misfolded proteorhodopsins, resulting from rho mutation, provoked intense protein aggregation culminating in UPR activation and consequent photoreceptor degeneration [84].

Thus, the cellular protein homeostasis in the photoreceptor is altered resulting in death cell mechanisms activation. In this process, the most common causes are (1) apoptosis, due to both caspase pathway activity and caspase-independent apoptosis, leading to oxidative stress augmentation and accumulation of oxidative DNA damage. However, in P23H and S334ter mutants was identified non-apoptotic markers of cell death, with suggestive indication of increased oxidative DNA damage (reviewed by [82].

Different cellular pathological processes are involved in RP photoreceptor degeneration, such as (2) ER disfunctions and consequent oxidative damage, demonstrated in the rhodopsin mutants T17M and E349X  where  increased pro-inflammatory markers were associated with ER-stress response. Photoreceptor programmed necrosis (3) could be mediated by receptor-interacting protein kinases (RIP), and inflammasome activation implicates in cell death in a P23H-1 rat model (reviewed by [82].”

Round 2

Reviewer 1 Report

Accept in present form